# Dichloroacetate reverses sepsis-induced hepatic metabolic dysfunction

Rabina Mainali[1†], Manal Zabalawi[2†], David Long[2], Nancy Buechler[1], Ellen Quillen[2], Chia-Chi Key[2], Xuewei Zhu[2], John S Parks[2], Cristina Furdui[2], Peter W Stacpoole[3], Jennifer Martinez[4], Charles E McCall[2]*, Matthew A Quinn[1,2]*

[1]Department of Pathology, Section on Comparative Medicine, Wake Forest School of Medicine, Winston-Salem, United States; [2]Department of Internal Medicine, Section on Molecular Medicine, Wake Forest School of Medicine, Winston-Salem, United States; [3]Division of Endocrinology, Diabetes and Metabolism, Department of Medicine and Department of Biochemistry and Molecular Biology, University of Florida College of Medicine, Gainesville, United States; [4]Immunity, Inflammation, and Disease Laboratory, National Institute of Environmental Health Sciences (NIEHS), National Institutes of Health (NIH), Research Triangle Park, Bethesda, United States

*For correspondence:
chmccall@wakehealth.edu
(CEMC);
mquinn@wakehealth.edu (MAQ)

†These authors contributed
equally to this work

Competing interests: The
authors declare that no
competing interests exist.

Reviewing editor: Zsolt Molnár,
University of Pécs, Medical
School, Hungary

**Abstract** Metabolic reprogramming between resistance and tolerance occurs within the immune system in response to sepsis. While metabolic tissues such as the liver are subjected to damage during sepsis, how their metabolic and energy reprogramming ensures survival is unclear. Employing comprehensive metabolomic, lipidomic, and transcriptional profiling in a mouse model of sepsis, we show that hepatocyte lipid metabolism, mitochondrial tricarboxylic acid (TCA) energetics, and redox balance are significantly reprogrammed after cecal ligation and puncture (CLP). We identify increases in TCA cycle metabolites citrate, cis-aconitate, and itaconate with reduced fumarate and triglyceride accumulation in septic hepatocytes. Transcriptomic analysis of liver tissue supports and extends the hepatocyte findings. Strikingly, the administration of the pyruvate dehydrogenase kinase (PDK) inhibitor dichloroacetate reverses dysregulated hepatocyte metabolism and mitochondrial dysfunction. In summary, our data indicate that sepsis promotes hepatic metabolic dysfunction and that targeting the mitochondrial PDC/PDK energy homeostat rebalances transcriptional and metabolic manifestations of sepsis within the liver.

## Introduction

Sepsis is a potentially life-threatening condition that occurs due to the body's overwhelming response to an infection. This reaction leads to aberrant immune responses, and if not diagnosed and treated early after its onset, it may limit survival by inducing coagulopathy, altered microvasculature, and dysregulation of the host's metabolism and organ function (*Lewis et al., 2016*; *Bermudes et al., 2018*; *Cheng et al., 2016*; *Levi and van der Poll, 2017*; *Schneck et al., 2017*; *McDonald et al., 2017*; *Kidokoro et al., 1996*). Sepsis accounts for one in three hospital deaths in the USA and millions of deaths each year globally, highlighting its hazard to public health (*Rhee et al., 2017*; *Buchman et al., 2020*). The high mortality rate associated with sepsis reflects the lack of clinically viable molecular-based therapeutic target. Therefore, understanding the pathogenesis of sepsis at both the molecular and organismal levels is of utmost importance to address major gaps in knowledge regarding sepsis.

The response to sepsis has classically been characterized as a biphasic phenomenon, where the acute phase is characterized by high energy consumption and hyper-inflammation associated with increased oxidation, followed by cellular reprogramming to a low-energy, anti-inflammatory state of

immunometabolic paralysis, with accompanying organ failure (*Rhee et al., 2017*; *Osuchowski et al., 2012*; *Chiswick et al., 2015*; *Xiao et al., 2006*). Additionally, during the early phase of sepsis, increased catabolism of fats, proteins, and carbohydrates, associated with high rates of oxygen consumption and ATP synthesis, is observed (*Singer, 2014*; *Druml et al., 2001*; *Mecatti et al., 2020*; *Seymour et al., 2013*). Following the acute phase is a hypometabolic state where ATP production and mitochondrial respiration decreases (*Singer, 2014*). It is postulated that this is a protective mechanism to overall lower the metabolic demands of the cell and help with its recovery (*Lewis et al., 2016*; *Quinones et al., 2014*; *Singer et al., 2004*).

Of particular interest, during the hyper-inflammatory anabolic phase of sepsis, an increase in the expression and activity of pyruvate dehydrogenase kinase 1 (PDK1) consistently occurs (*McCall et al., 2018*; *Vary, 1991*). This enzyme is one of four PDK isoforms that reversibly phosphorylates serine residues on pyruvate dehydrogenase complex (PDC) E1a subunit, inhibiting the conversion of pyruvate to acetyl coenzyme A (acetyl-CoA) (*Stacpoole, 1989*). Inhibition of this important enzyme central to glycolysis, tricarboxylic acid (TCA) cycle, oxidative phosphorylation (OXPHOS), and the lipogenic pathway is thought to be an important mechanism driving the dysfunction of mitochondrial respiration and cell bioenergetics observed during sepsis (*McCall et al., 2018*; *Zhu et al., 2020*). Therefore, PDK serves as a potential therapeutic target as its inhibition would allow for the downstream oxidation of glucose to continue, restoring OXPHOS and mitochondrial function known to be altered in both immune and non-immune cells during sepsis (*Seymour et al., 2013*; *McCall et al., 2018*; *Kelly and O'Neill, 2015*; *Eyenga et al., 2018*).

The liver is an important metabolic and immune organ due to its role in nutrient metabolism and production of acute phase proteins. However, our understanding of transcriptional alterations and subsequent metabolic phenotypes elicited by sepsis still remain limited (*Seymour et al., 2013*; *Ilaiwy et al., 2019*; *Dahn et al., 1995*; *Levy et al., 1968*; *Han et al., 2004*). Furthermore, the mechanisms by which dichloroacetate (DCA) elicits hepatic metabolic changes in response to sepsis are completely unknown. Hence, we set out to characterize the hepatic manifestations of sepsis, with the overall goal of identifying global metabolic pathways subject to dysregulation and whether these pathways are restored by DCA treatment.

## Results

### Sepsis impairs hepatic mitochondrial metabolism

To test the long-term hepatic transcriptional changes elicited by sepsis, we performed RNA-seq in whole livers 30 hr post-cecal ligation puncture (CLP). At this time point, septic mice exhibit tolerance, in which immunometabolic paralysis and end organ dysfunctions decrease survival. Ingenuity pathway analysis (IPA) of the top physiological pathways subject to alteration in response to sepsis revealed a significant increase in the acute phase response pathway, highlighting a persistent inflammatory state in the liver into the chronic phase of sepsis (*Figure 1a*). Of particular interest, OXPHOS and mitochondrial dysfunction were the top two enriched pathways in the liver of septic mice (*Figure 1a*). To gain further insight into the effects of sepsis on the transcriptional regulation of mitochondrial function, we performed gene set enrichment analysis (GSEA) of the OXPHOS pathway (GO:0006119). Septic mice at 30 hr decreased the transcriptional output of OXPHOS components as evidenced by a negative GSEA enrichment score (*Figure 1b*). Given the connection between OXPHOS and the TCA cycle, we next asked if polymicrobial infection would elicit similar alterations in this pathway. In accordance with the changes observed in the OXPHOS pathway, we found a negative enrichment score for TCA cycle enzymes (GO:0006099) in the liver of CLP mice (*Figure 1c*). Thus, our transcriptome data indicate that sepsis impairs mitochondrial metabolism in the liver.

Next, we wanted to assess whether the transcriptional changes elicited by sepsis would manifest in altered hepatic TCA cycle metabolism. Therefore, we performed global unbiased metabolomic screening in isolated hepatocytes from control and septic mice by ultrahigh-performance liquid chromotography–tandem mass spectroscopy (UPLC-MS/MS). In line with altered transcriptional regulation of the TCA cycle, we found that sepsis altered the relative abundance of multiple metabolites involved in the TCA cycle (*Figure 1d*). In particular, significant elevation of citrate and cis-aconitate was observed in septic hepatocytes at 30 hr (*Figure 1e,f*). Unlike macrophages, which shift their ratio of succinate and α-ketoglutarate to favor succinate accumulation over α-ketoglutarate

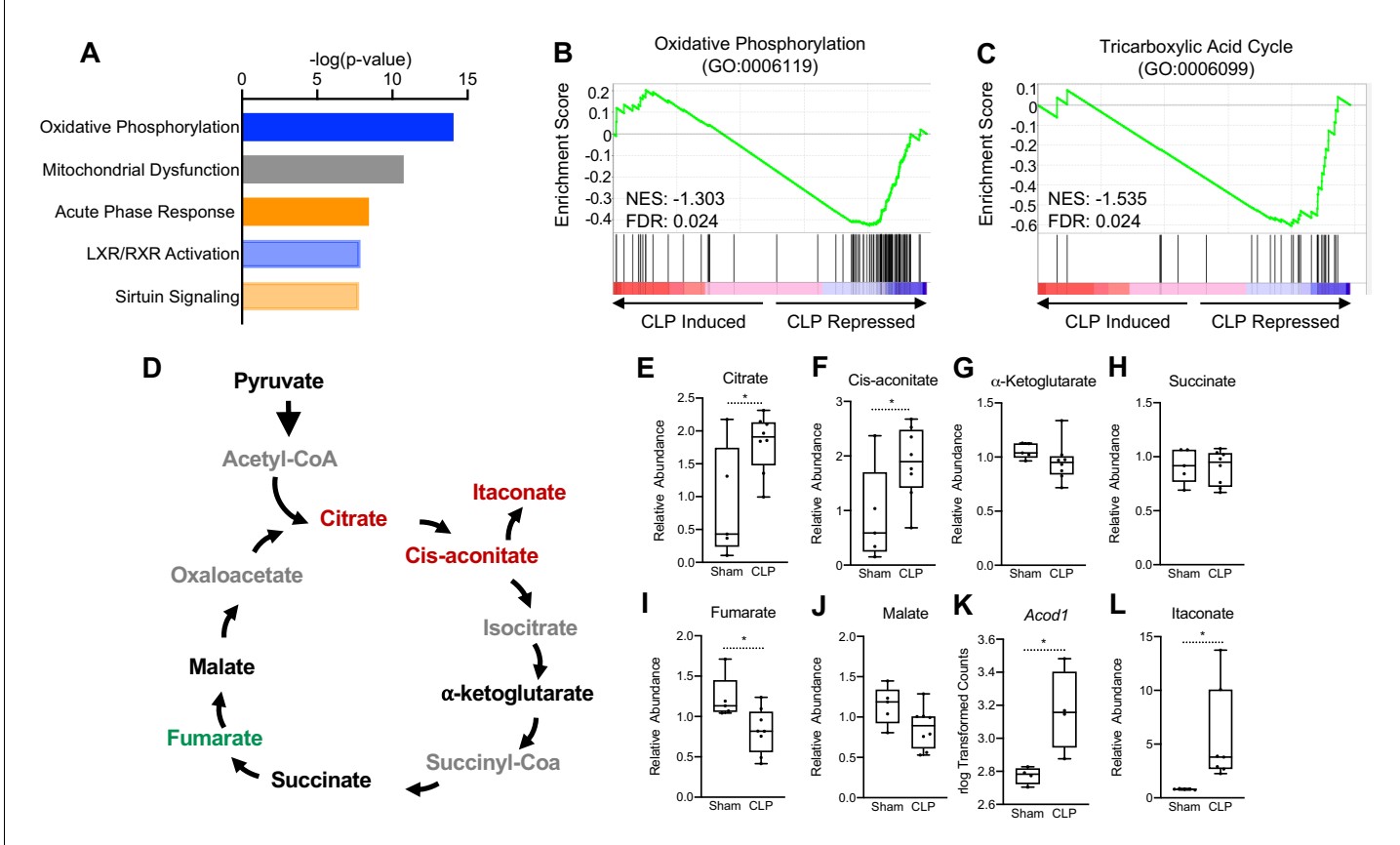

**Figure 1.** Sepsis Impairs hepatic mitochondrial metabolism. (A) Top five canonical pathways subject to transcriptional alterations in the liver identified by ingenuity pathway analysis of RNA-seq of sham versus CLP mice (n = 4 mice per group). Blue represents a negative z-score and orange represents a positive z-score. Shading indicates intensity of pathway activation/inhibition. (B) GSEA of the oxidative phosphorylation pathway (GO:0006119) indicating a negative normalized enrichment score (NES = −0.853). (C) GSEA of the TCA cycle pathway (GO:0006099) indicating a negative enrichment score (NES = −1.1956). (D) Schematic representation of hepatic TCA cycle metabolites altered during chronic sepsis. Red denotes a metabolite increased in response to sepsis; green indicates a metabolite decreased in response to sepsis; black indicates a metabolite unchanged in response to sepsis; gray indicates a metabolite not measured in our metabolomic screening. (E–J) Relative metabolite levels measured by ultrahigh-performance liquid chromatography–tandem mass spectroscopy (UPLC–MS/MS) from livers of sham and CLP mice 30 hr post-surgery (n = 5 sham; 8 CLP). (K) rlog transformed counts from RNA-seq of sham and CLP mice 30 hr post-surgery. (L) Relative itaconate levels in livers of sham and CLP mice measured by UPLC–MS/MS (n = 5 sham; 8 CLP). *p<0.05 as determined by Student's T-test.

(*Liu et al., 2017*; *Mills and O'Neill, 2014*; *Mills et al., 2016*), we found no changes in the levels of these metabolites in hepatocytes (*Figure 1G,H*). Furthermore, unlike macrophages (*Lampropoulou et al., 2016*), levels of fumarate decreased, but malate levels were unchanged in hepatocytes from septic animals (*Figure 1I,J*). Monocytes, macrophages, and dendritic cells reprogram the TCA cycle during acute inflammation to a catabolic tolerance phenotype by shunting cis-aconitate to itaconate through the enzymatic action of aconitase decarboxylase (ACOD1; also known as immune-responsive gene 1 [IRG1]) (*Lampropoulou et al., 2016*; *Liao et al., 2019*; *Mills et al., 2018*; *Daniels et al., 2019*). Accordingly, we also found that sepsis significantly induces *ACOD1* and elevates itaconate in isolate hepatocyte preparations (*Figure 1K,L*).

## Sepsis shifts the hepatic tedox balance toward oxidative stress

Perturbations in OXPHOS have been shown to promote the production of reactive oxygen species (ROS) (*Indo et al., 2007*). Furthermore, itaconate has been shown to induce activation of the master KEAP1/NRF2 antioxidant pathway leading to increased expression of redox regulatory enzymes in macrophages (*Mills et al., 2018*). Therefore, we examined if sepsis invokes a similar gene expression profile in hepatocytes. GSEA of ROS metabolism (GO:0072593) revealed a positive enrichment,

highlighting the transcriptional induction of genes involved in the regulation of redox state in livers of septic mice (*Figure 2a*). Given the induction of these genes, we wanted to determine whether sepsis alters key metabolites involved in the maintenance of cellular redox balance, focusing on the transsulfuration pathway leading to glutathione biosynthesis. Sepsis did not alter intracellular homocysteine levels in isolated hepatocytes (*Figure 2—figure supplement 1*); however, cystathionine decreased significantly (*Figure 2b*). Furthermore, we found a trend for decreased cysteine levels in the liver (p=0.0534) (*Figure 2c*) and significantly decreased glycine (*Figure 2d*). Most dramatically, sepsis depleted hepatocyte glutathione (*Figure 2e*) and led to decreased oxidized glutathione (GSSG) (*Figure 2f*). Ophthalmate has recently been shown to be a biomarker of oxidative stress and signals consumption of hepatic GSH (*Soga et al., 2006*). Functionally, we observe a significant accumulation of hepatic ophthalmate levels in response to sepsis, signifying a state of oxidative stress (*Figure 2g*).

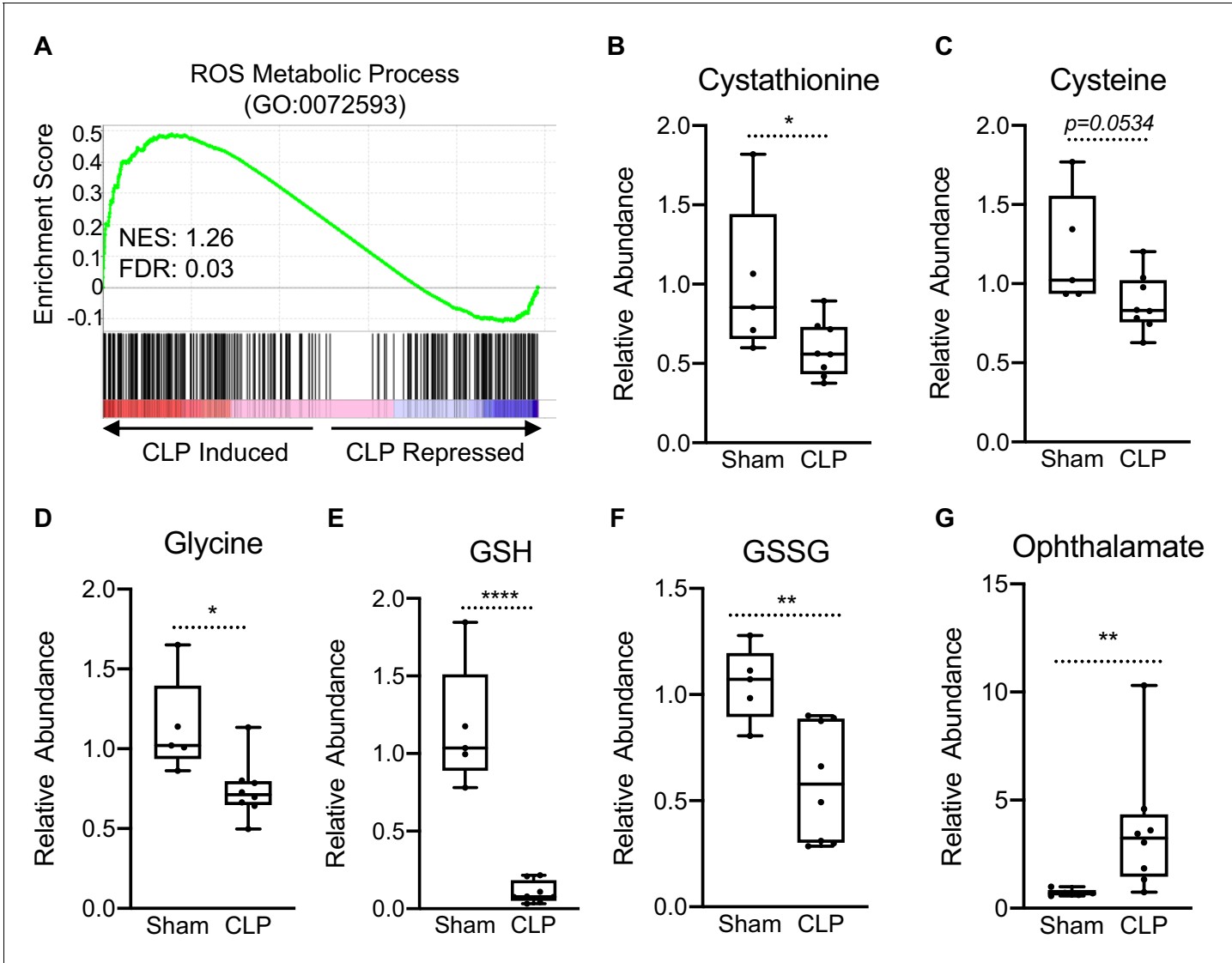

**Figure 2.** Impaired hepatic redox balance in septic mice. (**A**) GSEA of the ROS metabolic pathway (GO:0072593) from RNA-seq from livers comparing sham to CLP showing a positive enrichment score (NES = 1.117) (n = 4 mice per group). (**B–F**) Relative metabolite levels of metabolites involved in redox balance measured by UPLC–MS/MS (n = 5 sham; 8 CLP). (**G**) Relative levels of ophthalmate involved in redox balance measured by UPLC–MS/MS. *p<0.05, **p<0.01, ****p<0.0001 as determined by Student's T-test.

The online version of this article includes the following figure supplement(s) for figure 2:

**Figure supplement 1.** Hepatic homocysteine levels in sham and CLP mice.

## Sepsis promotes hepatic steatosis

Thus far, our data has revealed that sepsis induces significant impairments in hepatocyte OXPHOS, alterations in the TCA cycle, and induction of oxidative stress, findings that are also metabolic hallmarks of fatty liver disease (*Simões et al., 2018*). Because perturbations in global lipid profiles occur in septic patients (*Mecatti et al., 2020*), we evaluated the effects of CLP on the hepatic lipid metabolism pathway. In support of human data, our transcriptome data indicate significant repression of lipid metabolism in livers of septic mice (*Figure 3a*). Sepsis also decreased transcriptional output of fatty acid metabolic process components in the liver, as evidenced by a negative GSEA enrichment score (*Figure 3b*). In particular, we find alterations in genes involved in both fatty acid oxidation and fatty acid biosynthesis (*Figure 3—figure supplement 1*).

Since transcriptional regulation of hepatic lipid metabolism is significantly altered during sepsis, we further investigated the functional manifestation of the observed gene expression profile. Consistent with our transcriptional profiling studies, our unbiased metabolomic screening in hepatocytes also identified lipid metabolism as the top metabolic pathway altered in sepsis (*Figure 3c*). We observed increases in virtually all fatty acids and acylcarnitine derivatives assayed (*Figure 3d*). The dysregulation in hepatic lipid metabolism in response to sepsis ultimately culminates in the development of steatosis, as evidenced by increased free fatty acids, mono-, di-, and triglycerides (*Figure 3e*). Intriguingly, genes involved in fatty acid elongation and desaturation show impaired

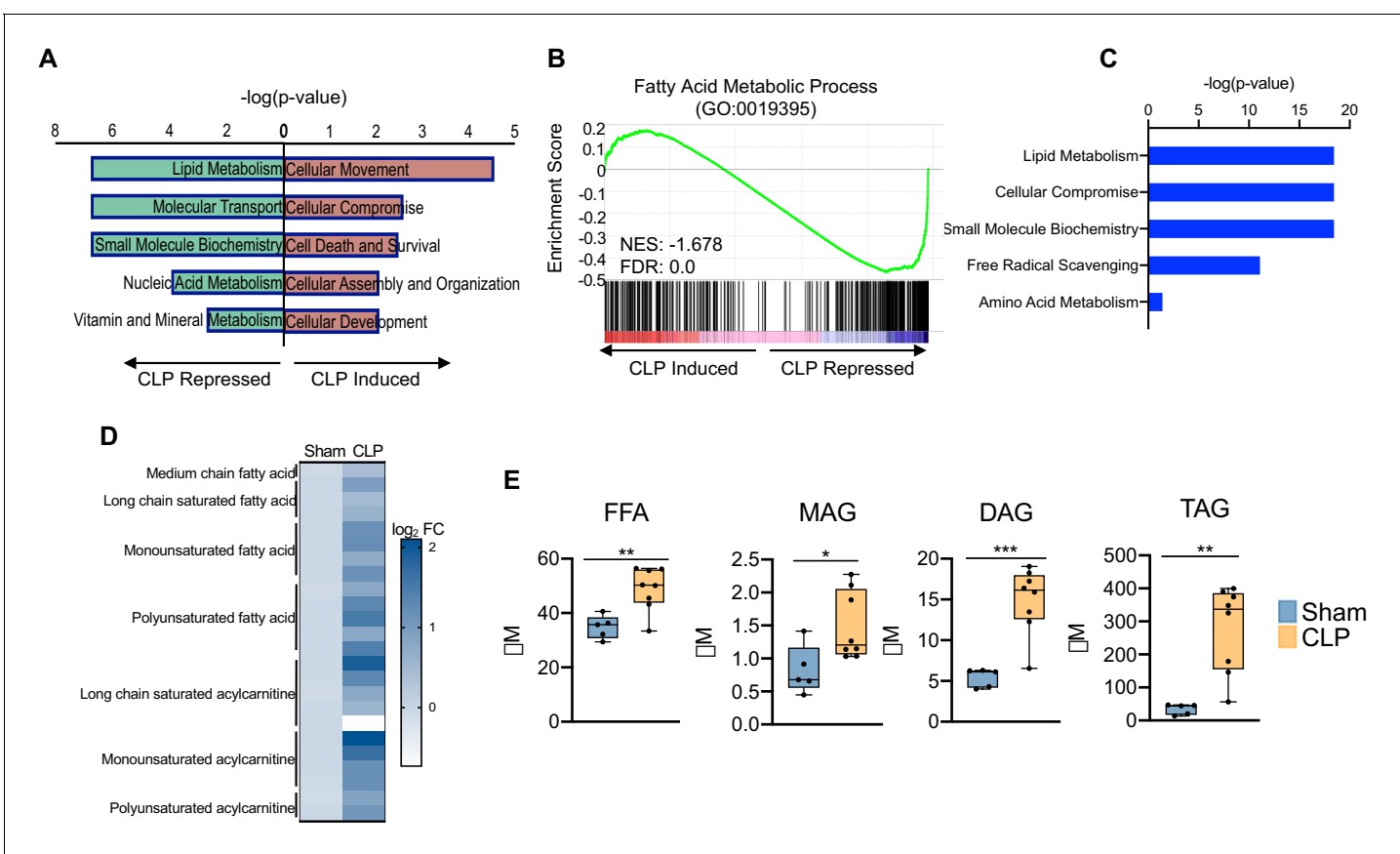

**Figure 3.** Sepsis promotes hepatic steatosis. (**A**) Top five induced and repressed physiological pathways in the liver of sham versus CLP mice identified by IPA of RNA-seq. (**B**) GSEA of the fatty acid metabolic pathway (GO:0019395) from RNA-seq from livers comparing sham to CLP showing a negative enrichment score (NES = −1.018) (n = 4 mice per group). (**C**) IPA of top five metabolic pathways significantly altered in the liver in response to sepsis identified by global metabolomic screening. (**D**) Heatmap representation of log2 fold change of different lipid species in sham and CLP mice measured by UPLC–MS/MS. (**E**) Quantification of lipids by UPLC–MS/MS in hepatocytes isolated from sham and CLP mice 30 hr post-surgery (n = 5 sham; n = 8 CLP). *p<0.05, **p<0.01, ***p<0.001 as determined by Student's T-test.

The online version of this article includes the following figure supplement(s) for figure 3:

**Figure supplement 1.** Fatty acid oxidation and biosynthetic pathways in septic livers.

expression in response to CLP at the time point surveyed (*Figure 3—figure supplement 1*), suggesting the metabolic changes observed at our time point potentially arise from earlier transcriptional changes and the observed repression is compensatory in nature.

## Sepsis remodels the hepatic lipidome

Given the severe dysregulation in hepatic lipid metabolism triggered by sepsis, we next sought to characterize the global consequences on hepatic lipid profiles. This was achieved by targeted lipidomic screening. Parallel to our unbiased metabolomic screening, we find sepsis leads to profound accumulation of almost all lipid species surveyed (*Figure 4a*). We next evaluated the hepatic lipid composition to discern which lipid species are most vulnerable to sepsis-induced alterations. We find very little alterations to the ceramide pool in septic hepatocytes (*Figure 4b*), with only modest changes in dihydroceramide (*Figure 4—figure supplement 1*). However, the hepatic fatty acid fractions displayed significant remodeling in response to sepsis (*Figure 4b*). Specifically, we find increases in free fatty acid species such as 16:0, 18:1, and 18:2 in septic hepatocytes (*Figure 4—figure supplement 2*). Moreover, hepatic phospholipid composition showed robust alterations following CLP (*Figure 4b*). Of the phospholipids (PLs) most significantly affected, we find accumulation of phosphotidylcholines (PCs), phosphotidylethanolamine (PE), and sphingomyelin (SM) (*Figure 4c*). Of the hepatic PC pool, we find the most abundant species (16:0/18:2 and 16:0/18:1) to show increased accumulation in response to sepsis (*Figure 4—figure supplement 2*). We also observed significant increases in the most abundant PE species (18:0/20:4, 16:0/22:6, and 18:0/22:6) and SM species (22:0, 16:0, and 20:0) in septic hepatocytes (*Figure 4—figure supplement 2*). One phospholipid species, lysophosphatidylcholine, displayed a decrease in concentration during sepsis; however, the hepatic concentration of this PL species is significantly lower than other PL species assayed (*Figure 4c*). Finally, we detected no differences in other PL species such as phosphotidylinositol (PI) and lysophosphatidylethanolamine (LPE) and no differences in hepatic cholesterol ester levels (*Figure 4—figure supplement 3*). Collectively, our data detail the impact of sepsis on hepatic lipid composition and robustly show a global rise in nearly all lipid species.

## PDK inhibition attenuates sepsis-induced transcriptional and metabolic changes in the liver

We have previously reported that PDK inhibition by the pyruvate analog and pan-PDK inhibitor DCA promotes PDC-dependent immunometabolic adaptations to sepsis and increases survival (*McCall et al., 2018*). Given the establishment of hepatocyte mitochondrial dysfunction and steatosis during sepsis, we asked whether pharmacological targeting of the hepatic PDK/PDC axis would mitigate the disruption of key metabolic and bioenergetic processes induced by sepsis. To test this postulate, we administered DCA to septic mice 24 hr after CLP onset and assessed the global transcription profile of liver 6 hr later via RNA-seq. DCA treatment of septic mice reversed the majority of sepsis-regulated gene networks (*Figure 5a*). Most notably, pathways involved in redox metabolism, lipid metabolism, and mitochondrial dysfunction are returned to basal levels with the administration of DCA in septic mice (*Figure 5b*).

Given our previous results indicating the transcriptional changes associated with sepsis in the liver underlie functional alterations in metabolism, we next sought to determine whether the DCA reversal of transcriptional changes leads to reprogramming of metabolism to a pre-sepsis state. Our RNA-seq data showed a positive enrichment score for production of ROS along with negative enrichment for glutathione conjugation. Strikingly, the depletion of redox components cystathionine, cysteine, glycine, hypotaurine, taurine, and glutathione elicited by sepsis was completely reversed by administration of DCA (*Figure 5c*). More importantly, we observed a significant decrease in hepatic ophthalmate levels, signifying a decrease in oxidative stress (*Figure 5c*). The restoration of key redox metabolites in septic mice following DCA administration prompted us to investigate the mitochondrial consequences of this pharmacological intervention. With the exception of fumarate, DCA completely restores TCA metabolite levels to that of control levels (*Figure 5d*). Finally, we wished to define the effects of DCA treatment on the hepatic lipidome. Lipidomic analysis shows a protective effect of DCA in ameliorating sepsis-induced steatosis. In summary, these data reveal pharmacological inhibition of the PDK pathway restores transcriptional and metabolic changes associated with sepsis in the liver.

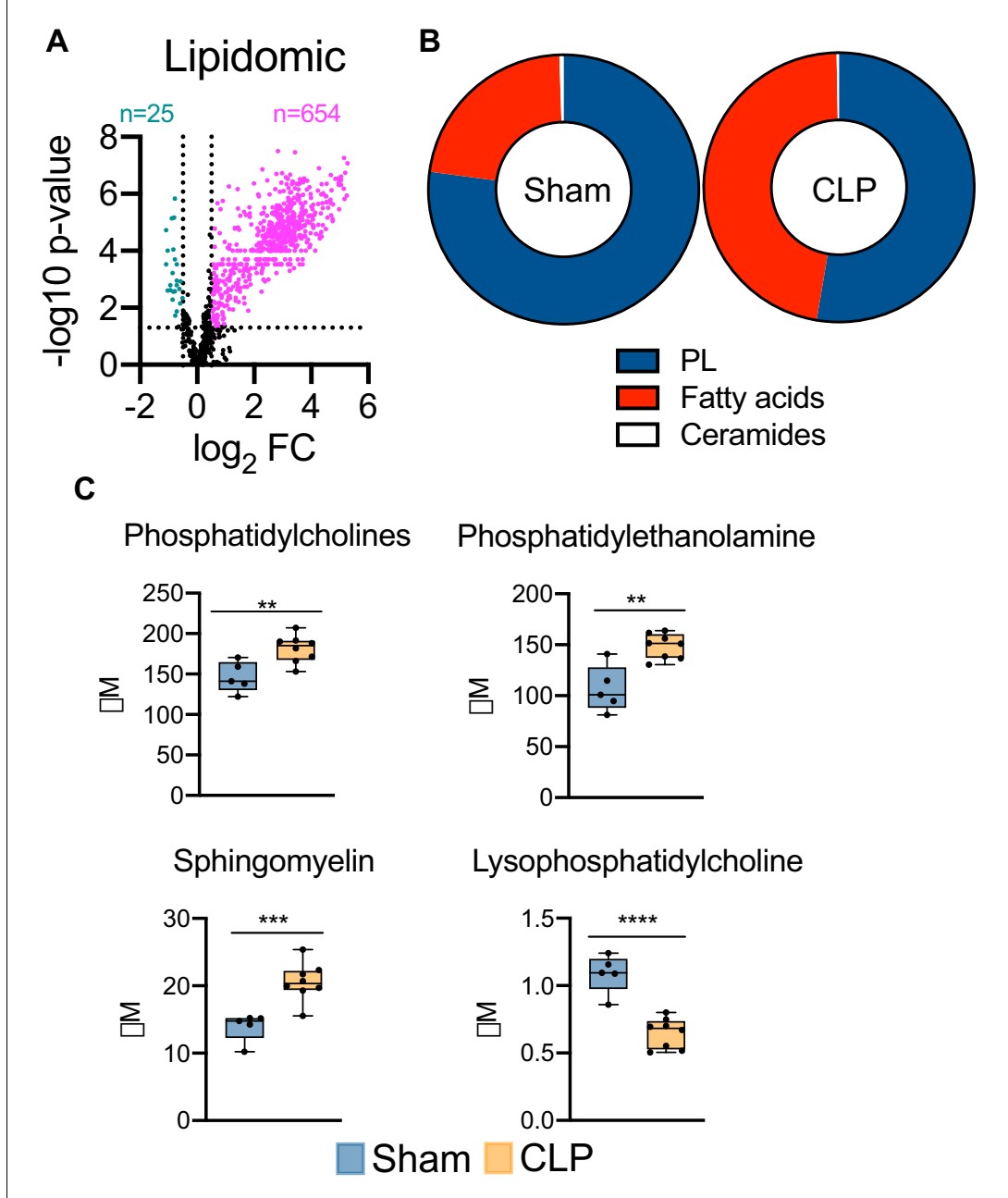

**Figure 4.** Sepsis reprograms the hepatic lipidome. (A) Volcano plot of significantly altered lipids from sham and CLP hepatocytes measured by UPLC–MS/MS (n = 5 sham; n = 8 CLP). (B) Pie chart of lipid compositions from sham and CLP mice. (C) Quantification of lipids by UPLC–MS/MS in hepatocytes isolated from sham and CLP mice 30 hr post-surgery (n = 5 sham; n = 8 CLP). **p<0.01, ***p<0.001, ****p<0.0001 as determined by Student's T-test.

The online version of this article includes the following figure supplement(s) for figure 4:

**Figure supplement 1.** Hepatic ceramide levels in sham and CLP mice.

**Figure supplement 2.** Phosphatidylcholine, phosphatidylethanolamine, and sphingomyelin quantifications.

**Figure supplement 3.** PI, LPE, and CE levels in septic livers.

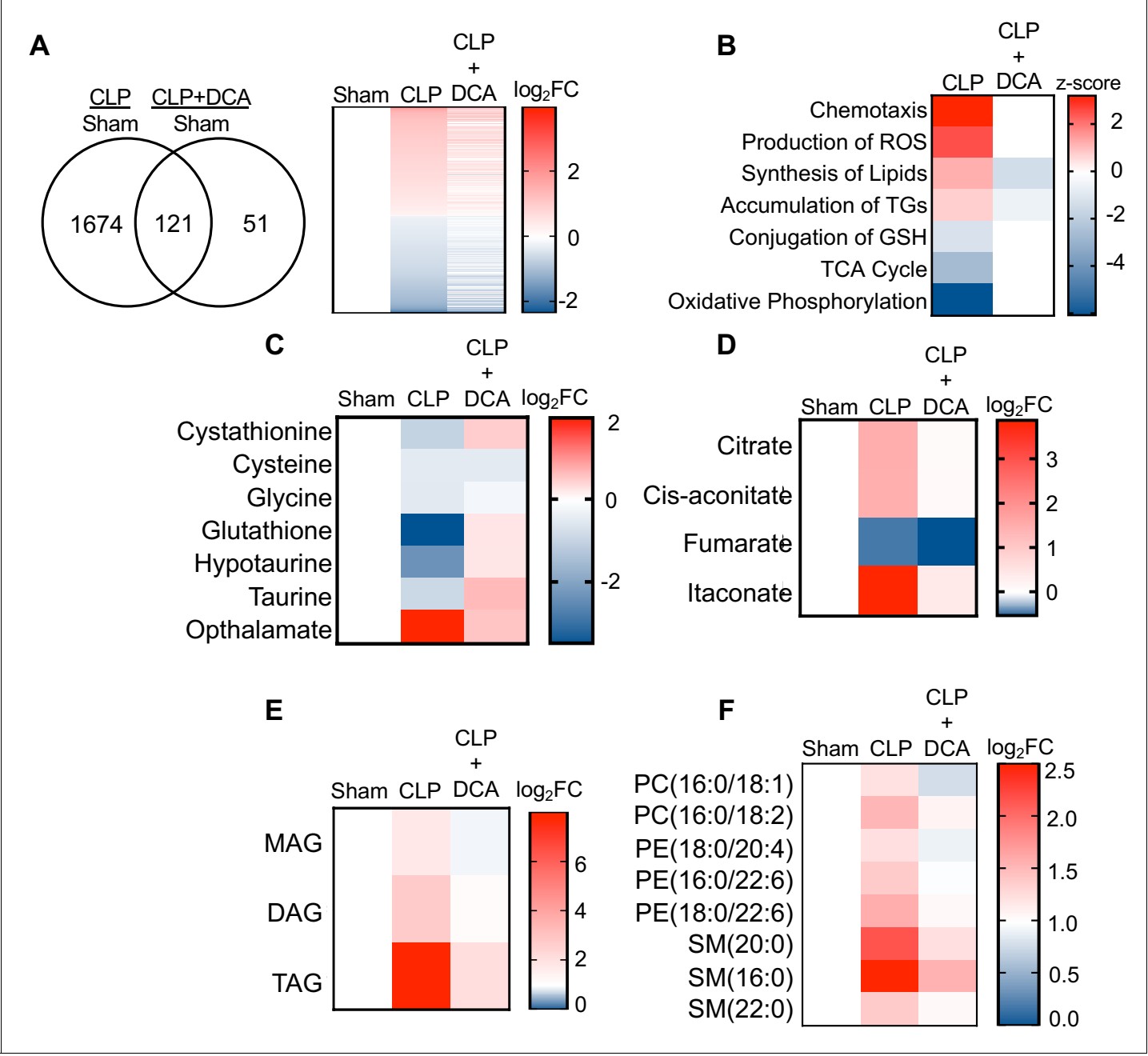

**Figure 5.** PDK inhibition restores hepatic metabolism in septic mice. (A) Venn diagram of differentially expressed genes (DEGs) assessed by RNA-seq in CLP versus sham compared to CLP+DCA versus sham 30 hr post-surgery (left). Heatmap of average log2 fold change of DEGs in sham, CLP, and CLP +DCA (right) (n = 4 mice per group). (B) Heatmap depiction of z-scores of top canonical pathways identified by IPA in CLP versus sham and CLP+DCA versus sham. (C) Heatmap depiction of average log2 fold change in metabolite levels involved in redox balance in sham, CLP, and CLP+DCA 24 hr post-surgery measured by UPLC–MS/MS (n = 5 sham; n = 8 CLP and CLP+DCA). (D) Heatmap depiction of average log2 fold change in metabolite levels involved in TCA cycle in sham, CLP, and CLP+DCA 30 hr post-surgery measured by UPLC–MS/MS (n = 5 sham; n = 8 CLP and CLP+DCA). (E) Heatmap depiction of average log2 fold change in MAG, DAG, and TAG levels in sham, CLP, and CLP+DCA 30 hr post-surgery measured by UPLC–MS/MS (n = 5 sham; n = 8 CLP and CLP+DCA). (F) Heatmap depiction of average log2 fold change in phospholipids in sham, CLP and CLP+DCA 30 hr post-surgery measured by UPLC–MS/MS (n = 5 sham; n = 8 CLP and CLP+DCA).

## Discussion

The failure of over 50 trials of anti-inflammatory and anti-cytokine agents for treatment of severe sepsis has led to a reevaluation of the concept of sepsis as causing death by unbridedled inflammation (*Abraham, 1999*). Instead, it has become increasingly clear that mitochondrial bioenergetics and metabolic reprogramming play vital roles in organ and immune cell dysfunction during sepsis that may lead to sustained organ failure and immunometabolic paralysis (*Chang and Pearce, 2016*; *Patel and Powell, 2017*). Consequently, much attention has been focused recently on the immuno-metabolic consequences of inflammation in immune cells (*Buck et al., 2017*). However, significant gaps exist in our understanding of the metabolic adaptations induced by systemic inflammation from sepsis in vital organs. Addressing this limitation is critical because sepsis survival depends on restoring both organ and immune cell homeostasis following dysregulated and disseminated inflammation (*Singer et al., 2016*). In the present study, we reveal severe disruption of several key hepatic metabolic pathways in septic mice, including TCA cycle activity, OXPHOS, redox, and lipid metabolism during the prolonged tolerant phase of sepsis. Importantly, DCA administration restored TCA cycle flux, ameliorated oxidative stress, and reversed sepsis-induced steatosis.

Remodeling of TCA cycle flux is well appreciated to support immune cell effector function in the face of inflammatory challenges (*Cheng et al., 2016*; *Mills et al., 2016*; *Williams and O'Neill, 2018*; *Zasłona and O'Neill, 2020*). For example, succinate primes inflammation through succinate dehydrogenase (SDH)-mediated ROS generation and ATP synthesis (*Mills et al., 2016*; *Williams and O'Neill, 2018*). SDH-derived ROS enhances IL-1β production by activating HIF-1α and activating the NLRP3 inflammasome (*Vary, 1991*). Inhibition of SDH activity increases IL-10 production and skews inflamed immune cells toward an anti-inflammatory response (*Mills et al., 2016*). Fumarate, on the other hand, possesses anti-inflammatory properties as it inhibits pro-inflammatory cytokine production either by activating the KEAP1/NRF2 pathway or via inhibiting pathways like NF-κB and MAPK that lie downstream of TLR signaling in immune cells (*McGuire et al., 2016*; *Giustina et al., 2018*). Our findings extend the concept that the TCA cycle is subjected to reprogramming in response to sepsis in non-immune cells such as hepatocytes. Furthermore, we found similarities as well as differences in terms of TCA cycle remodeling in hepatocytes compared to immune cell effects, clearly indicating this pathway is regulated in a cell-type- and context-dependent manner during sepsis.

In response to lipopolysaccharide (LPS) stimulation, macrophages significantly upregulate IRG1 and accumulate itaconate from the decarboxylation of cis-aconitate (*Lampropoulou et al., 2016*; *Mills et al., 2018*). One of the primary functions itaconate serves in immune cells is to limit IL-1 β production through direct inhibition of SDH and activation of the KEAP1/NRF2 and IκBζ-ATF3 pathways (*Lampropoulou et al., 2016*; *Mills et al., 2018*; *Bambouskova et al., 2018*). In line with what occurs in macrophages, we observed an increase in IRG1 and itaconate in septic hepatocytes at 30 hr. While we did not fully characterize the hepatic function of itaconate, a recent paper demonstrates its anti-inflammatory effects during ischemia-reperfusion injury in the liver (*Yi et al., 2020*). The deletion of IRG1 heightens inflammation and liver damage and renders hepatocytes susceptible to oxidative injury after I/R injury. Furthermore, itaconate administration reduces liver damage and inflammation associated with I/R in IRG1 KO mice, also emphasizing its anti-inflammatory and hepatoprotective effects (*Yi et al., 2020*). Given the immunomodulatory effects of itaconate and its accumulation in the liver in our study, we hypothesize that itaconate directs hepatocyte shifts in both TCA cycle metabolism and mitochondrial energetics. Moreover, immune cells have been shown to depend on itaconate accumulation to confer tolerance (*Domínguez-Andrés et al., 2019*). Our similar findings within the liver during the prolonged phase of sepsis suggests this organ may utilize similar mechanisms to promote tissue tolerance during chronic sepsis. Intriguingly, DCA reduces itaconate levels, which we also find in a human monocyte model of sepsis (*Zhu et al., 2020*).

Similar to immune cells, liver cells increase ROS production during sepsis (*Giustina et al., 2019*; *Chen et al., 2019*). We offer a potential mechanism whereby sepsis leads to significant depletion of redox metabolites within the glutathione pathway ultimately culminating in loss of cellular glutathione levels resulting in ophthalmate accumulation in hepatocytes.

Sepsis profoundly alters lipid metabolism, resulting in significant increases in plasma fatty acid and glycerol concentrations, changes reported to predict prognosis in septic patients (*Langley et al., 2013*). Furthermore, patients who died from sepsis had evidence of hepatic steatosis affecting 5–80% of liver parenchyma (*Koskinas et al., 2008*). Human data supports a robust hepatic

steatosis phenotype subsequent to sepsis. One vital lipid metabolic pathway that can contribute to steatosis is de novo lipogenesis. While we did not directly measure hepatic lipogenesis, our metabolomics screening did show an increase in hepatic citrate and free fatty acid levels suggesting a increased lipogenesis during sepsis. Functionally, immune cells exploit the lipogenic pathway for the generation of malonyl-CoA and subsequent malonylation of glycolytic enzymes, which ultimately contributes to the metabolic adaptation of macrophages to inflammatory cues (*Galván-Peña et al., 2019*). While this work did not investigate the mechanisms and factors driving alterations in hepatic lipid metabolism during sepsis, we hypothesize altered regulation of transcription factor peroxisome proliferator-activated receptor alpha (PPARα) to be play a critical role. PPARα is considered to be one of the key regulators of fatty acid oxidation as it controls the expression of numerous genes in this pathway upon activation by unsaturated fatty acids (*Georgiadi and Kersten, 2012*). Multiple studies have reported altered expression of PPARα during sepsis (*Paumelle et al., 2019*; *Standage et al., 2012*; *Standage et al., 2017*). Of interest, recently Van Wyngene et al. reported the biological activity of PPARα to be significantly lowered in a CLP model of sepsis (*Van Wyngene et al., 2020*). This resulted in impaired fatty acid uptake and β-oxidation, driving the increase of fatty acids levels in circulation and accumulation of lipid droplets in the liver (*Van Wyngene et al., 2020*). Furthermore, Wyange et al. also demonstrated that activation of PPARα improved hepatic lipid metabolism and survival during sepsis, highlighting the protective nature of this pathway during sepsis (*Van Wyngene et al., 2020*).

In addition to accumulation of free fatty acids and triglycerides in the liver, our lipidomic analysis revealed significant increases in PLs. In the present study, we find hepatic levels of the PLs phosphatidylcholine, phosphatidylethanolamine, and sphingomyelin to be heightened during sepsis, indicating abnormal metabolism of PLs. PC and PE are the most abundant PLs that make up cellular membranes and an abnormally high ratio of PC and PE has been reported to influence energy metabolism and linked to fatty liver disease and impairment in liver regeneration after injury (*Kennelly et al., 2017*; *Ling et al., 2012*; *Li et al., 2006*). Altered PC:PE ratio has also been shown to influence the dynamics and regulation of lipid droplets contributing toward steatosis, a phenotype we also report in our study (*Kennelly et al., 2017*; *Ling et al., 2012*; *Listenberger et al., 2018*). The dysregulation of hepatic PL metabolism may be a contributing mechanism underlying steatosis, which is a topic of future investigation. Additionally, PLs serve as precursor molecules to bioactive lipids, which are involved in numerous signal transduction cascades (*Sunshine and Iruela-Arispe, 2017*). Investigating whether signaling properties of bioactive lipids in the liver during sepsis is pathogenic or contributes to sepsis resolution is also needed in the future.

PDC is a master metabolic regulator controlling the conversion of pyruvate to acetyl-CoA in the mitochondria (*Stacpoole, 2012*) and its inactivation contributes to the metabolic reprogramming that occurs in immune cells in response to inflammatory signaling (*Zhu et al., 2020*; *Stacpoole, 2012*). PDK is a negative regulator of the PDC, as it phosphorylates PDC and inhibits the conversion of pyruvate to acetyl-CoA (*Stacpoole, 2012*). During sepsis, the expression and activity of PDK1 in immune cells is heightened, contributing to the dysfunction of mitochondrial metabolism. One mechanism driving this observed increase is through pro-inflammatory mediators such as LPS and interferon gamma (IFN-γ) (*Min et al., 2019*). Another possible mechanism contributing to PDK activation in the context of sepsis is through glucocorticoid signaling (*Huang et al., 2002*). In fact, starvation is well established to activate the PDK pathway (*Hutson and Randle, 1978*; *Baxter and Coore, 1978*; *Fuller and Randle, 1984*; *Denyer et al., 1986*; *Wu et al., 2000*). Based on our findings of elevated glucocorticoid levels in both the circulation and the liver, we hypothesize the stress hormone pathway as a contributor to PDK activation during sepsis.

We reported in a sepsis monocyte model that DCA reduced TCA cycle catabolic effects concurrent with increasing amino acid anaplerotic catabolism of branched-chain amino acids, leading to increased TCA-driven anabolic energetics (*Zhu et al., 2020*). In the present study, we find restoration of hepatic TCA metabolites, decreased triglyceride accumulation, lessening of lipid synthesis, and ammelioration of oxidative stress in septic mice after DCA administration. Overall, we demonstrate that hepatic transcriptional and metabolic dysfunction improves after targeting the PDK/PDC axis with DCA.

Notably, our interpretation of results is based on a single 30 hr time point, which was selected based on our previous publication showing reversal of disease tolerance mechanisms and increased survival following DCA intervention at 24 hr (*McCall et al., 2018*). Indeed, the metabolic changes at

30 hr in this study align with repressed mitochondrial respiration and extracellular acidification reported in this earlier publication. Future time course studies will take advantage of this data to expand our knowledge of hepatic metabolic perturbations during the sepsis response and provide higher granularity of the regulatory events underlying the DCA effects. Furthermore, while our current work provides a comprehensive atlas of hepatic metabolic and transcriptional pathways subject to dysregulation during prolonged sepsis, acute metabolic manifestations on a global scale remain unknown. Indeed, it is well documented that early metabolic changes are induced in the immune compartment in response to pathogen exposure, which ultimately shape the host response to infection including induction of inflammation as well as establishment of immune tolerance (*van der Poll et al., 2017*). Previous work from our group demonstrates extreme metabolic flexibility that occurs in a coordinated fashion in myeloid cells (*Zhu et al., 2019*), opening the possibility that the liver may behave in a similar dynamic fashion. Moreover, prior work has demonstrated rearrangements of certain hepatic metabolic pathways at the transcriptional level within hours of sepsis induction (*Recknagel et al., 2012*). For example, amino acid, fatty acid, and redox pathways are downregulated transcriptionally in the liver within 6 hr of sepsis (*Recknagel et al., 2012*). Intriguingly, we find similar alterations at the metabolite level during prolonged sepsis, indicating early transcriptional changes elicited by sepsis may prime the liver for the later metabolic manifestations observed in our current study. Future work evaluating the contribution of these early transcriptional changes to acute metabolic reprogramming in hepatocytes during the resistance phase is needed. These studies will illuminate whether the findings reported here represent protective mechanisms associated specifically with tissue tolerance or are pathogenic in nature, a product of aberrant response to pathogenic exposure.

In terms of molecular mechanisms underlying the hepatic metabolic manifestations of sepsis, several potential candidates exists. Firstly, the mammallian target of rapamycin (mTOR) could be contributing to sepsis-induced metabolic disturbances in the liver. The foundation for this hypothesis is due to mTOR being demonstrated to be an essential component of the inflammatory response as well as underlying the glycolytic switch observed during acute sepsis (*Cheng et al., 2016*; *Cheng et al., 2014*). The metabolic signature observed here during the prolonged phase of sepsis is characterized by decreased mitochondrial and glycolytic derived energy, consistent with a state of impaired mTOR activity. Evaluating the effects of sepsis on altering hepatic mTOR signaling will shed light on whether there is acute activation of this pathway during the resistance phase and whether it shifts toward inhibition during the tolerant phase. Opposite to the anabolic mTOR pathway is the AMP-activated kinase (AMPK) pathway, which promotes catabolism. AMPK is known to play a critical role in maintaining cellular energy homeostasis and its activation is important in the restoration of metabolic balance. Numerous studies utilizing animal models such as acute lung injury, ischemia-reperfusion, acute kidney injury, and hemorrhagic shock have shown protective effects of AMPK, its activation decreasing organ injury and inflammation (*Zhao et al., 2008*; *Peralta et al., 2001*; *Howell et al., 2013*). Of interest, activation of AMPK during sepsis minimizes liver and renal endothelial damage by decreasing inflammatory cytokine production (*Escobar et al., 2015*). Similarly, using transgenic mouse models, others have shown that deletion of AMPKα in myeloid cells exacerbates polymicrobial sepsis by promoting the release of HMGB1 and endotoxic shock, while activation of AMPK had protective effects (*Huang et al., 2018*). Overall, given the evidence, it is clear that activation of the AMPK signaling serves to limit inflammation and overall organ dysfunction. However, the exact mechanism by which AMPK signaling is dysregulated and the impact on metabolic remodeling in the liver during sepsis needs further investigation.

While we did not directly test here the physiological/pathophysiological contribution of individual metabolic pathways during the course of sepsis, we hypothesize that the overwhelming hepatic metabolic response during prolonged sepsis is pathogenic in nature. Our interpretation is based on our DCA results. We have previously demonstrated that DCA treatment reverses immunoparalysis and promotes host survival (*McCall et al., 2018*). Our results indicate that the vast majority of metabolic changes observed during non-lethal sepsis are completely reversed in DCA-treated mice. We surmise that if these metabolic pathways were contributing to host survival, the DCA treatment would have enhanced their activity (i.e. DCA would increase metabolites increased in response to sepsis). Furthermore, we postulate that the observed hepatic metabolic changes associated with prolonged sepsis may in fact be directly contributing to immunoparalysis and subsequent host demise. Further work dissecting the contribution of these hepatic metabolic changes during the course of non-lethal

sepsis is needed to establish the physiological/pathophysiological contribution in the context or pathogen resistance and tissue tolerance.

Alternatively, it is possible that some of the metabolic changes associated with sepsis in the liver confer host protection and tissue resolution. For example, we observed a significant decrease in hepatic fumarate levels 30 hr post-sepsis and an even further reduction in the DCA-treated septic mice. We speculate that the decrease in endogenous hepatic fumarate may lend itself to organ protection and sepsis resolution. Fumarate derivatives however have been shown to possess anti-inflammatory, antioxidant, and pro-survival properties in the liver during sepsis (*Shalmani et al., 2018*). Whether endogenous fumarate posseses differential effects than its derivatives (dimethyl fumarate) is unknown. Other TCA cycle derivatives such as dimethyl itaconate and 4-octyl itaconate have shown to exert differential effects than the endogenous form (*Swain et al., 2020*). Future work dissecting the role of endogenous fumarate and derivatives during lethal and non-lethal sepsis is warranted to illuminate whether this metabolite promotes tissue damage and whether the reduction is compensatory in nature.

In particular, further investigation specifically into the PDK pathway to determine which isoform underlies the hepatic manifestions of sepsis is needed. This is warranted given multiple isoforms of PDK expressed in the liver (*Klyuyeva et al., 2019*) and DCA inhibits multiple isoforms of PDK (*Bao et al., 2004*; *Pratt and Roche, 1979*). The results of this study fill a gap in understanding how sepsis at the molecular level alters liver metabolism and expand the current concepts on physiology and pathophysiology of resistance and tolerance of hepatocyte during sepsis (*Weis et al., 2017*; *Soares et al., 2017*). It is still unclear from our studies however if the effects of DCA on reversing hepatic metabolism are due to its effects on ameliorating immunoparalysis and enhancing pathogen clearance.

Here, we have confirmed and extended our previous findings that stimulation of mitochondrial OXPHOS metabolism by DCA in murine sepsis may be therapeutic in ameliorating the sepsis-induced immunometabolic paralysis of liver and potentially other vital organs. While encouraging, we recognize the limitations of extrapolating such positive findings in murine models to septic patients. Nevertheless, intravenous DCA has been used safely in critically ill patients (*Stacpoole et al., 1983*; *Stacpoole et al., 1992*; *Stacpoole et al., 1988*), supporting the translational potential of the current and related investigations. Furthermore, our data demonstrate that the reversal of organ metabolic dysfunction may be a potential mechanism by which DCA confers protection in murine models of sepsis.

## Materials and methods

### Animal experiments

Male C57BL/6J mice aged 8–10 weeks were purchased from The Jackson Laboratory (Bar Harbor, ME). All animals were subjected to a 12:12 hr dark/light cycle with ad libitum access to standard rodent chow and water. CLP model was used to induce sepsis as previously described (*McCall et al., 2018*; *Vachharajani et al., 2014*). Mice were randomly assigned to our experimental groups. Briefly, cecum was ligated and punctured two times with a 22-gauge needle. Contents were then returned, and incision was closed in two layers (peritoneum and skin). Sham operation where abdominal incision was made, but cecum not ligated or punctured was used as a control. Subcutaneous fluids (1 ml normal saline) were given to each animal. Mice were euthanized 30 hr post-surgery for tissue collection. Dichloroacetate (DCA) (Sigma; MO, USA) was administered (25 mg/kg) intraperitoneally at 24 hr post-surgery and tissues collected 6 hr post DCA administration (30 hr post-surgery). No explicit power analysis was used to calculate sample size. Our sample size was based on consultation with Metabolon for metabolomic screening.

### Hepatocyte Isolation

Hepatocytes were isolated via portal vein perfusion and collagenase digestion as previously described (*Chen et al., 2000*). Following perfusion, liver cells were liberated by gentle dissociation in Dulbecco's modified Eagle's medium (DMEM) (ThermoFisher; CA, USA). Cells were then filtered through nylon mesh to remove cellular debris and connective tissue and resulting cells pelleted by

centrifugation at 50 g for 1 min. After three washes with DMEM, cells were counted and viability assessed via Trypan Blue exclusion.

## RNA-sequencing

RNA was isolated from whole liver tissues using Trizol and the RNeasy RNA isolation kit (Qiagen; MD, USA) according to manufacturer's protocol. One microgram of high-quality RNA (RIN > 8) was used as a template for library generation using the Illumina TruSeq RNA Sample Prep Kit v2 (Illumina; CA, USA) according to the manufacturer's protocol. Generated libraries were then poly(A) enriched for mRNA prior to sequencing. Indexed samples were sequenced at 100 bp paired-end protocol with the NovaSeq 6000 (Illumina), generating approximately 20–30 million reads per sample. Sequenced reads were aligned to the University of California Santa Cruz (UCSC) mm10 reference genome using STAR v2.5 as previously described (*Dobin et al., 2013*). The mapped read counts were quantified by Subread featureCounts v1.5.0-p1 (*Liao et al., 2014*). Differentially expressed genes (DEGs) were determined by DESeq2 v1.14.1 (*Liao et al., 2014*) using a false discovery rate of 0.05. Ingenuity pathway analysis (Qiagen) and Gene Set Enrichment Analysis v4.0.3 (GSEA) were further used as previously described (*Quinn et al., 2018*). Sequencing was performed one time with four biological replicates per group.

## Ultrahigh-performance liquid chromatography–tandem mass spectroscopy

Hepatocytes were isolated described above for metabolomic screening via ultrahigh-performance liquid chromotography–tandem mass spectroscopy (UPLC–MS/MS) (Metabolon; NC, USA). Briefly, 150–200 µl cell pellets per animal were used as starting material. Samples were prepared using the automated MicroLab STAR system (Hamilton; NV, USA). Proteins were precipitated using methanol under vigorous shaking for 2 min followed by centrifugation. Prior to analysis organic solvents were removed with TurboVap (Zymark; MA, USA) and overnight storage under nitrogen. Dried samples were reconstituted with solvents compatible with the three following analytical methods: (1) reverse phase (RP)/UPLC–MS/MS methods with positive-ion mode electrospray ionization (ESI), (2) RP/UPLC–MS/MS with negative-ion mode ESI, and (3) HILIC/UPLC–MS/MS with negative-ion mode ESI. Resulting samples were analyzed with the ACQUITY UPLC (Waters; MA, USA) and a Q-Exactive high-resolution/accurate mass spectrometer (ThermoScientific; MA, USA) interfaced with a heated electrospray ionization (HESI-II) source and Orbitrap mass analyzer operated at 35,000 mass resolution. An acidic positive-ion condition was used on an aliquot optimized to detect more hydrophilic compounds. Another acidic positive-ion condition was ran but chromotographically optimized for hydrophobic compounds. Basic negative-ion optimized conditions were also used on a separate C18 column. Resulting raw data was extracted and peaks identified using Metabolon's hardware and software. Compounds were identified by comparing to known library entries of purified standards or recurrent unknown entities. A library of authenticated standards contain retention time/index (RI), mass to charge ratio (m/z), and chromotographic data (MS/MS spectral data) on all library compounds. Three criteria are used to identify chemicals: (1) RI within a narrow window of proposed identification, (2) accurate mass match to the library ±10 ppm, and (3) MS/MS forward and reverse scores between the experimental data and authentic standards. Peaks were quantified using area under the curve. For complex lipid panel, lipids were extracted with methanol:dichloromethane in the presence of internal standards. Extracts were concentrated under nitrogen and reconstituted in 250 ml of 10 mM ammonium acetate dichloromethane:methanol (50:50). Mass spectrometry analysis was performed in a Shimazdu LC with nano PEEK tubing and the Sciex SlexIon-5500 QTRAP (Sciex; MA, USA). Both negative and positive mode electrospray were used. Individual lipid species were quantified by taking the peak area ratios of target compounds and their assigned internal standards then multiplying by the concentration of added internal standards. Lipid class concentrations were calculated from the sum of all molecular species within a class, and fatty acid compositions were determined by calculating the proportion of each class comprised by individual fatty acids. Mass spectrometry analysis was performed one time.

## Acknowledgements

The authors of this manuscript would like to thank the National Institute of Environmental Health Sciences Epigenomics and DNA Sequencing Core for their help with RNA-sequencing and Dr. Sara Grimm for her analysis of transcriptome data. This work was supported by NIH Intramural Program 1ZIAES10328601 (JM), NIH R01 HL132035 (XZ), NIH R01 HL119962 (JSP), NIH K01 DK117069 (CK), NIH K01 AG056663 (EQ), NIH R01 AI065791 (CEM), R01 GM102497 (CEM), and R35 GM126922 (CEM). The authors would like to also acknowledge intellectual support provided by the Center for Redox Biology and Medicine at Wake Forest School of Medicine.

## Additional information

### Funding

| Funder | Grant reference number | Author |
| --- | --- | --- |
| National Institute of Environmental Health Sciences | 1ZIAES10328601 | Jennifer Martinez |
| National Heart, Lung, and Blood Institute | R01 HL132035 | Xuewei Zhu |
| National Institute of Diabetes and Digestive and Kidney Diseases | K01 DK117069 | Chia-Chi Key |
| National Institute on Aging | K01 AG056663 | Ellen Quillen |
| National Institute of Allergy and Infectious Diseases | R01 AI065791 | Charles E McCall |
| National Institute of General Medical Sciences | R01 GM102497 | Charles E McCall |
| National Institute of General Medical Sciences | R35 GM126922 | Charles E McCall |
| National Heart, Lung, and Blood Institute | R01 HL119962 | John S Parks |

The funders had no role in study design, data collection and interpretation, or the decision to submit the work for publication.

### Author contributions

Rabina Mainali, Data curation, Formal analysis, Investigation, Writing - original draft; Manal Zabalawi, Data curation, Formal analysis, Investigation, Methodology, Writing - review and editing; David Long, Data curation, Formal analysis, Investigation, Writing - review and editing; Nancy Buechler, Data curation, Investigation; Ellen Quillen, Formal analysis, Visualization, Writing - original draft; Chia-Chi Key, Formal analysis, Investigation, Writing - original draft; Xuewei Zhu, Writing - review and editing; John S Parks, Cristina Furdui, Formal analysis, Writing - review and editing; Peter W Stacpoole, Writing - original draft; Jennifer Martinez, Data curation, Formal analysis, Writing - original draft; Charles E McCall, Supervision, Funding acquisition, Writing - original draft, Project administration; Matthew A Quinn, Conceptualization, Data curation, Formal analysis, Supervision, Investigation, Writing - original draft, Project administration

### Author ORCIDs

Chia-Chi Key http://orcid.org/0000-0003-0669-2936
John S Parks http://orcid.org/0000-0002-5227-8915
Matthew A Quinn https://orcid.org/0000-0002-3528-6569

### Ethics

Animal experimentation: This study was performed in strict accordance with the recommendations in the Guide for the Care and Use of Laboratory Animals of the National Institutes of Health. All of

the animals were handled according to approved institutional animal care and use committee (IACUC) protocols (A19-097) Wake Forest School of Medicine.

## Decision letter and Author response

Decision letter https://doi.org/10.7554/eLife.64611.sa1
Author response https://doi.org/10.7554/eLife.64611.sa2

## Additional files

### Supplementary files
- Source data 1. Lipidomic analysis.
- Source data 2. Metabolomic analysis.
- Source data 3. Transcriptomic analysis.
- Transparent reporting form

### Data availability

Sequencing data have been deposited in GEO under accession code GSE167127.

The following dataset was generated:

| Author(s) | Year | Dataset title | Dataset URL | Database and Identifier |
|-----------|------|---------------|-------------|-------------------------|
| Quinn MA, Martinez J | 2021 | Effects of sepsis and dichloroacetate on hepatic transcriptome | https://www.ncbi.nlm.nih.gov/geo/query/acc.cgi?acc=GSE167127 | NCBI Gene Expression Omnibus, GSE167127 |

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
