## [Decision Letter]

**Acceptance summary:**

All reviewers found this study interesting and also important. It is potentially of broad interest to the readers of *eLife* and also a valuable addition to the field of immunometabolism. It provides a valuable dataset detailing hepatic metabolic programs induced by well-established septic model. It also contributes to the growing interest in understanding the complex bi-directional biology of inflammation-induced metabolic changes on an organismal level and whether this understanding can lead to novel therapeutic strategies to diseases of pathologic systemic inflammation.

**Decision letter after peer review:**

Thank you for submitting your article " reverses sepsis-induced hepatic metabolic dysfunction" for consideration by *eLife*. Your article has been reviewed by three peer reviewers, and the evaluation has been overseen by a Reviewing Editor and Jos van der Meer as the Senior Editor. The following individuals involved in review of your submission have agreed to reveal their identity: Michael Bauer (Reviewer #1); Andrew Wang (Reviewer #2).

The reviewers have discussed the reviews with one another and the Reviewing Editor has drafted this decision to help you prepare a revised submission.

Summary:

Sepsis related organ dysfunction is life threatening condition, which is one of the leading causes of death in hospitalized patients. Despite major achievements in cardiovascular, respiratory and renal support the mortality especially related to septic shock remains high, around 50%. Liver is one of the vital organs but unfortunately at present we can only affect its function indirectly on the one hand, and on the other hand liver support devices are far behind as far as efficacy is concerned when compared to that of used in hemodynamics, mechanical ventilation or renal replacement therapy. Therefore, better understanding of the liver in sepsis could be a major step forward. For these reasons the authors aim to perform this experiment should be congratulated.

Their aim was to test the long-term hepatic transcriptional changes, for which they designed an animal experiment using the CLP model compared to sham controls. In addition to performing RNA-seq in whole livers, they also investigated whether transcriptional changes manifest in altered hepatic TCA cycle metabolism.

They found that in septic mice, 30 hours after CLP, hepatocyte lipid metabolism, mitochondrial TCA energetics and redox balance were significantly reprogramed.

Based on the group's earlier findings that pyruvate dehydrogenase kinase (PDK) inhibition by the pyruvate analogue dyhydroacetate (DCA) promotes immunometabolic adaptations to sepsis and increased survival, they also tested the effects of DCA administration in the current experiment. They found similar results, namely that DCA reversed dysregulated hepatocyte metabolism and mitochondrial dysfunction.

Essential revisions:

We appreciate the high-quality work of the authors, and after careful evaluation our experts raised several concerns, comments and questions, such as:

– The importance of metabolic changes and the signalling process within the observation period of 30 hours is missing from the full picture.

– The overall impact of the current findings on the outcome is also unclear.

– It is also unclear if the observed changes in liver metabolism are adaptive or maladaptive in sepsis.

– The link between hepatic damage and mortality in the experiment is also difficult to prove – although the authors suggest this as primary cause of sepsis related death.

– The shift towards the accumulation of free fatty acids in the experiment also requires further explanations.

– There are no "magic bullets" in sepsis therapy and DCA is certainly difficult to believe being the one – every trial which tested therapies targeting one molecule or process in sepsis and showed promises in the animal laboratory, failed to show clinical benefit (in fact often caused harm).

– Furthermore, the prerequisite of recovery from septic shock is the rapid rebalancing in the DO2/VO2 ratio to avoid or correct tissue hypoperfusion and oxygen debt – hence, pharmacological interventions cannot be separated from the global resuscitation process.

– Therefore, the clinical relevance of the findings of the current experiment are questionable and renders the need of revision of the Discussion and the conclusions.

For more details, please see the reviewers' comments.

Reviewer #1:

Mainali et al. present a comprehensive analysis of metabolic changes in the liver in Response to CLP in mice. The investigations include metabolomics, lipidomics and transcriptomics as well as mitochondrial and specific redox parameters. The study reveals a pronounced sepsis induced shift of hepatocyte metabolism and a salvage effect of the PDK inhibitor DCA on increased mortality of septic mice and concomitant metabolic traits. In spite of the restricted novelty of most of the specific findings the reviewer suggests to consider acceptance of the manuscript in *eLife* due to the solidity, comprehensibility and coherency of the data. Metabolic changes in the course of sepsis are currently in the focus of the research field and drugs affecting these crucial metabolic changes during sepsis are promising candidates for successful treatment of this fatal syndrome. Before acceptance of the manuscript a revision along the comments attached is required.

The Discussion part of the manuscript is fixed on the interpretation of metabolic changes happening 30 h after CLP surgery. The authors do not touch metabolic changes until (and after) 30 h. Such considerations might be important for embedding the given data in the current concept on physiological and pathophysiological resistance and tolerance responses of immune cells and parenchyma in sepsis. Hence, to reduce its descriptive character the authors might considerably expand the Discussion.

A similar restrictive interpretation is also observable for signaling processes controlling metabolism. How are mTOR and AMPK as key mediators of anabolic and catabolic processes involved in the control of the metabolic events observed after sepsis induction? Key references describing this issue like Cheng et al., 2016, are missing in the given manuscript.

Reviewer #2:

In this study, Mainali and Zabalawi et al. find that significant hepatic metabolic reprogramming occurs in the CLP model of sepsis. Using -omics approaches in ex vivo hepatocytes, they show CLP-induced reprogramming of the hepatic TCA cycle, lipid metabolism, and redox balance. Finally, in continuation of McCall et al., 2018, they show that PDK inhibition, which they previously showed improved CLP survival outcomes, reverses many of these CLP-induced hepatic changes. This is a very well-done study that supports the growing body of work that immunometabolic reprogramming is critical for organ preservation and survival during systemic inflammation.

My major reservation in this study is that the authors do not demonstrate the relevance of the presented observations to the overall impact on sepsis physiology. That is, it is not surprising that hepatocytes, like most cells, undergo a lot of reprogramming during sepsis. Are these observed hepatic changes adaptive and promote organ preservation and survival, or are they maladaptive? It is also not clear that hepatic damage (or damage of "metabolic tissues") is the "primary cause of sepsis death". Along these lines, in Figure 5, the reference to the protective effects of PDK inhibition in CLP-driven mortality is difficult to interpret because, in the cited study, the authors clearly show that PDK inhibition given in the same regimen as the current study improves mortality primarily by reversing immunoparalysis and enhancing pathogen clearance. It remains unclear how hepatic reprogramming contributes to this mechanism.

Thus, as presented, their findings do not support their main argument that hepatic reprogramming impacts survival in sepsis (Abstract) in either direction. In my opinion, it also does not add, from a pre-clinical perspective, to the potential utility of DCA in treating sepsis beyond their 2018 study.

Reviewer #3:

1) The data show a significant shift towards accumulation of free fatty acids and especially 16:0, 18:1 and 18:2 in addition to a significant increase in MUFAs and PUFAs in CLP vs. Sham. I had only one question concerning this result: Can the authors provide an explanation for this result shown in Figure 3D and Figure 4—figure supplement 2A? Are the enzymes, normally responsible for elongation and desaturation e.g. ELOVL and SCD respectively, altered under septic conditions?

2) As useful information for the research community, it would be helpful if the authors could discuss the altered lipid metabolism in hepatic parenchymal cells (induced lipogenesis, reduced FFA-oxidation) shown in the manuscript within the Discussion section.

---

## [Author Response]

Essential revisions:We appreciate the high-quality work of the authors, and after careful evaluation our experts raised several concerns, comments and questions, such as:– The importance of metabolic changes and the signalling process within the observation period of 30 hours is missing from the full picture.

This important point was raised by reviewer #1. Please see below for details on how we updated our Discussion to touch on acute changes during sepsis that were not covered by our experimental approach.

– The overall impact of the current findings on the outcome is also unclear.

This point was raised by reviewer #2. We have updated our Discussion to include our interpretation of our findings, particularly in regard to DCA, in context of sepsis treatment and pathogenesis.

– It is also unclear if the observed changes in liver metabolism are adaptive or maladaptive in sepsis.

This concern was raised by reviewer #2. Please see below for details on how we have updated our Discussion to discuss our interpretation in terms of whether the changes observed are adaptive or pathogenic.

– The link between hepatic damage and mortality in the experiment is also difficult to prove – although the authors suggest this as primary cause of sepsis related death.

We agree with this comment and have updated our Discussion to soften our conclusions regarding hepatic damage as causal of sepsis related death.

– The shift towards the accumulation of free fatty acids in the experiment also requires further explanations.

This point was raised by reviewer #3. We have updated our Results and Discussion to address this. Please see below for details.

– There are no "magic bullets" in sepsis therapy and DCA is certainly difficult to believe being the one – every trial which tested therapies targeting one molecule or process in sepsis and showed promises in the animal laboratory, failed to show clinical benefit (in fact often caused harm).

We agree with this comment that there is no magic bullet to sepsis therapy. However, we have added references in which DCA treatment has been used to treat lactic acidosis associated with sepsis. See below for details.

Reviewer #1:Mainali et al. present a comprehensive analysis of metabolic changes in the liver in Response to CLP in mice. The investigations include metabolomics, lipidomics and transcriptomics as well as mitochondrial and specific redox parameters. The study reveals a pronounced sepsis induced shift of hepatocyte metabolism and a salvage effect of the PDK inhibitor DCA on increased mortality of septic mice and concomitant metabolic traits. In spite of the restricted novelty of most of the specific findings the reviewer suggests to consider acceptance of the manuscript in eLife due to the solidity, comprehensibility and coherency of the data. Metabolic changes in the course of sepsis are currently in the focus of the research field and drugs affecting these crucial metabolic changes during sepsis are promising candidates for successful treatment of this fatal syndrome. Before acceptance of the manuscript a revision along the comments attached is required.The Discussion part of the manuscript is fixed on the interpretation of metabolic changes happening 30 h after CLP surgery. The authors do not touch metabolic changes until (and after) 30 h. Such considerations might be important for embedding the given data in the current concept on physiological and pathophysiological resistance and tolerance responses of immune cells and parenchyma in sepsis. Hence, to reduce its descriptive character the authors might considerably expand the Discussion.

We appreciate this insightful comment from the reviewer. We agree that by choosing one time point at 30 hours post-sepsis does not convey a complete picture in relation to metabolic reprogramming undertaken by the liver during sepsis. While we agree, we do believe that our comprehensive metabolomic, lipidomic and transcriptomic profiling is a first step to understanding how vital organs like the liver are altered in response to sepsis. In an effort to provide a more comprehensive interpretation of our results in the context of literature we have significantly updated our Discussion. In particular, we have included new discussion points relating to acute changes associated with metabolic reprogramming and the effects on pathogen resistance and tissue tolerance. We have also included additional references incorporating previous literature demonstrating transcriptional changes in metabolic pathways dysregulated in the liver during acute sepsis. Moreover, we updated our Discussion to reflect our itaconate findings in the context of tissue tolerance and immunoparalysis based on previous reports.

A similar restrictive interpretation is also observable for signaling processes controlling metabolism. How are mTOR and AMPK as key mediators of anabolic and catabolic processes involved in the control of the metabolic events observed after sepsis induction? Key references describing this issue like Cheng et al., 2016, are missing in the given manuscript.

We agree with the reviewer that our interpretation of our results in the context of the previous literature was limited. To address this, we have incorporated an additional Discussion point regarding the potential role of mTOR and AMPK signaling in the regulation of metabolism during sepsis. Furthermore, we have made sure to include seminal publications such as Cheng et al. in our Discussion, particularly regarding tissue tolerance and immunoparalysis.

Reviewer #2:[…] My major reservation in this study is that the authors do not demonstrate the relevance of the presented observations to the overall impact on sepsis physiology. That is, it is not surprising that hepatocytes, like most cells, undergo a lot of reprogramming during sepsis. Are these observed hepatic changes adaptive and promote organ preservation and survival, or are they maladaptive?

We agree with the reviewer that this is a primary limitation to our study. We believe however, that obtaining molecular insight into whether these changes are maladaptive or protective would require substantial experiments that would be outside the scope of the current manuscript. To address this however, we have updated our Discussion to hypothesize our interpretation of our results in regulating host survival or demise.

It is also not clear that hepatic damage (or damage of "metabolic tissues") is the "primary cause of sepsis death". Along these lines, in Figure 5, the reference to the protective effects of PDK inhibition in CLP-driven mortality is difficult to interpret because, in the cited study, the authors clearly show that PDK inhibition given in the same regimen as the current study improves mortality primarily by reversing immunoparalysis and enhancing pathogen clearance. It remains unclear how hepatic reprogramming contributes to this mechanism.

This is a very important point raised by the reviewer. We have subsequently softened our interpretation of clinical efficacy and therapeutic potential and have taken out statements indicating that hepatic damage is the primary cause of sepsis-related death. Furthermore, we have updated our Discussion to broaden our interpretation of our DCA effects in the context of our previous work. We pose the hypothesis that the effects of DCA in modulating immunoparalysis could in part be due to its effects on restoring hepatic metabolic function. This opens the possibility that hepatic metabolism could be a direct regulator of immunoparalysis in the context of sepsis and warrants future investigation.

Thus, as presented, their findings do not support their main argument that hepatic reprogramming impacts survival in sepsis (Abstract) in either direction. In my opinion, it also does not add, from a pre-clinical perspective, to the potential utility of DCA in treating sepsis beyond their 2018 study.

We agree that our main argument that hepatic reprogramming impacts sepsis survival is premature to make. To rectify this, we have removed that concept from our manuscript. We have expanded our DCA findings to indicate the potential utility of targeting the PDK axis may be mediated through hepatic reprogramming. Please refer to the Discussion section.

Reviewer #3:1) The data show a significant shift towards accumulation of free fatty acids and especially 16:0, 18:1 and 18:2 in addition to a significant increase in MUFAs and PUFAs in CLP vs. Sham. I had only one question concerning this result: Can the authors provide an explanation for this result shown in Figure 3D and Figure 4—figure supplement 2A? Are the enzymes, normally responsible for elongation and desaturation e.g. ELOVL and SCD respectively, altered under septic conditions?

We believe this to be an interesting hepatic phenomenon observed during sepsis. We have subsequently revisited our RNA-seq analysis of sham and CLP animals from 30h post-surgery and evaluated the expression of several enzymes involved in elongation and desaturation. We found repressed expression of all enzymes assayed and have offered our interpretation of this data within the Results. Furthermore, we have added additional data to Figure 3—figure supplement 1 panel B.

2) As useful information for the research community, it would be helpful if the authors could discuss the altered lipid metabolism in hepatic parenchymal cells (induced lipogenesis, reduced FFA-oxidation) shown in the manuscript within the Discussion section.

We appreciate this comment from the reviewer. We have added Discussion points regarding hepatic lipid metabolism in the context of sepsis. Furthermore, we have added discussion surrounding the major lipid sensor PPARa and what is known about its role during sepsis and its potential contribution to our observed phenotype.